# Earliest geometries: A cognitive investigation of Howiesons Poort engraved ostrich eggshells

**Valentina Decembrini**[ID][1], **Ludovica Ottaviano**[1], **Mattia Cartolano**[1],
**Enza Elena Spinapolice**[2], **Silvia Ferrara**[ID][1]*

**1** Department of Classical Philology and Italian Studies, University of Bologna, Bologna, Italy,
**2** Department of Ancient World Studies, Sapienza University of Rome, Rome, Italy

* s.ferrara@unibo.it

## Abstract

This article presents the first quantitative geometric and spatial analysis of engraved ostrich eggshell (EOES) fragments from the Howiesons Poort (HP) technocomplex of the African late Middle Stone Age (MSA), to evaluate whether the EOES demonstrates genuine formal structuring and visuo-spatial organization. By considering their 'non-accidental properties'—such as curvature, parallelism, and co-termination—which remain consistent across different viewpoints, as well as their metric properties, including angular inclinations, based on empirical thresholds, we show that the HP dataset systematically employs salient geometric features. These features are combined and embedded through complex cognitive operations, including the iteration and alignment of parallel lines, rotation of lines generating intersections with variable angular openings, and translation of specific elements nested within organized spatial layouts. These engravings therefore constitute an early material expression of complex graphic representation, attesting to a species-specific human capacity for organizing geometric thought. Overall, the patterns reflect a system of rules through which *Homo sapiens* in the HP organized visual forms, revealing the cognitive foundations of structured graphic behavior.

## Introduction

The Howiesons Poort (HP) technocomplex of the late Middle Stone Age (MSA) in southern Africa has yielded the earliest evidence of engraved ostrich eggshell (EOES) fragments. The EOES findings from three HP sites — Klipdrift and Diepkloof rock shelters in South Africa, and Apollo 11 in Namibia — constitute an abundant assemblage characterized by incised lines arranged in recurrent organizations [1–5]. In this paper, we aim to investigate the cognitive processes underlying the construction of these patterns through the first method-based, quantitative assessment of their formal and measurable features. Indeed, these engravings have being classified into motif groups according to their shared 'geometric (*sensu lato*)' characteristics,

**Data availability statement:** "The relevant data is available at the following link: https://doi.org/10.5281/zenodo.17953360".

**Funding:** This study was funded by the FIS Advanced Grant, project SAPIENCE Symbols, Preliteracy and Code Evolution, Ministry of Research Italy, Ministero dell'Istruzione, dell'Università e della Ricerca (FIS-2023-00821 to Prof Silvia Ferrara).

**Competing interests:** The authors have declared that no competing interests exist.

based on direct visual inspection of the fragments [2]. But, to date, no dedicated study has examined statistically their formal geometry, which is a crucial aspect for understanding whether these patterns reflect genuine cognitive structuring.

The EOES have typically been discussed within the broader debate on behavioral modernity [6,7] and frequently included within the same symbolic 'package' as other early manifestations of abstract marking and symbolic material culture [8–11]. However, rather than focusing on their symbolic potential and interpretation, here we delve into a deeper analytical level, exploring the cognitive foundations that underlie the construction of the engravings.

Our analysis builds on studies suggesting that flexible Euclidean intuitions are part of human core knowledge as a cross-cultural universal, together with spatial orientation and object recognition [12–18]. These intuitions include Euclidean geometric features, such as line, point, parallelism, and right angle, and are defined as 'conceptual primitives of geometry' [12]. Importantly, these primitives can be combined through a set of rule-based operations which can be iterative (insertion of additional elements within the same hierarchical level) or embedded (nested elements on additional hierarchical levels) [17–19].

Even if some Euclidean principles are ingrained in our cognitive system, they are occasionally violated in human perception [14]. Investigations on cognitive maps have found that our spatial thought rather rests on a hybrid Euclidean and non-Euclidean spatial representation [20]. Nonetheless, this hybrid space can be successfully projected onto a Euclidean (2D) surface – and vice versa – thanks to a slight spontaneous distortion that aims to keep visual salience (the perceptual prominence or recognizability) of basic geometric features such as right angles and parallelism [21–24]. This perspective is crucial for our analysis, as it facilitates an investigation of the EOES through measurable geometric parameters.

To this end, we retrace the engravings and segment them into their basic components to extract key geometric variables. We calculate the geometric attributes and assess whether these patterns reflect cognitive iterative mechanisms, embedding strategies, and visuo-spatial awareness and planning. The results show a high degree of regularity in the arrangement of salient geometric features, involving procedural and hierarchical operations. What will become apparent is, thus, the emergence of a 'geometric grammar', that is a system of rules through which *Homo sapiens* in the HP organized visual forms, revealing the cognitive foundations of structured graphic behavior.

## Materials and methods

### Contextualizing the EOES

Ostrich eggshells (OES) have been used as water flasks ever since the African Middle Stone Age and are still used by present-day foragers in southern Africa [25,26]. During the late phase of the MSA, in southern and eastern Africa, the archaeological record shows the earliest attestations of OES engravings. The oldest EOES are found in the southern African HP technocomplex. Although its chronology is still debated, due to disagreements on dating Diepkloof's stratigraphy [27,28], we

here adopt dates ranging roughly between ~65 and ~60 ka (64.8 ka and 59.5 ka [29]), which are more homogenous with the other HP sites. EOES were found specifically in three sites: Diepkloof and Klipdrift in South Africa, and Apollo 11 in southern Namibia (Fig 1). All three sites display the OES engravings, across the duration of the HP sequence (ca. 5 ka; Table 1), thus making them a continuous, consistent feature. After a hiatus, they reappear later in the Late Stone Age, with similar features [30].

The EOES have been classified into distinct motifs [2–4]. Accordingly, some site-specific and chronological preferences can be observed (Fig 2). To cite some examples, the diamond-shaped motif is only seen in Kilpdrift during the final HP phase, the orthogonal hatched band occurs exclusively in the intermediate and final phases in Diepkloof, and in Apollo 11 only the sub-parallel intersecting lines motif is present, while totally absent in Klipdrift.

However, we take the assemblage in its entirety, since the aim of this analysis is to study the cognitive underpinnings of the EOES: this approach allows us to capture overarching structural regularities. Methodologically, this choice is supported by the material, cultural, geographic, and chronological uniformity of the HP, which ensures that our comprehensive analysis is not hindered by the selective criteria of the available sources.

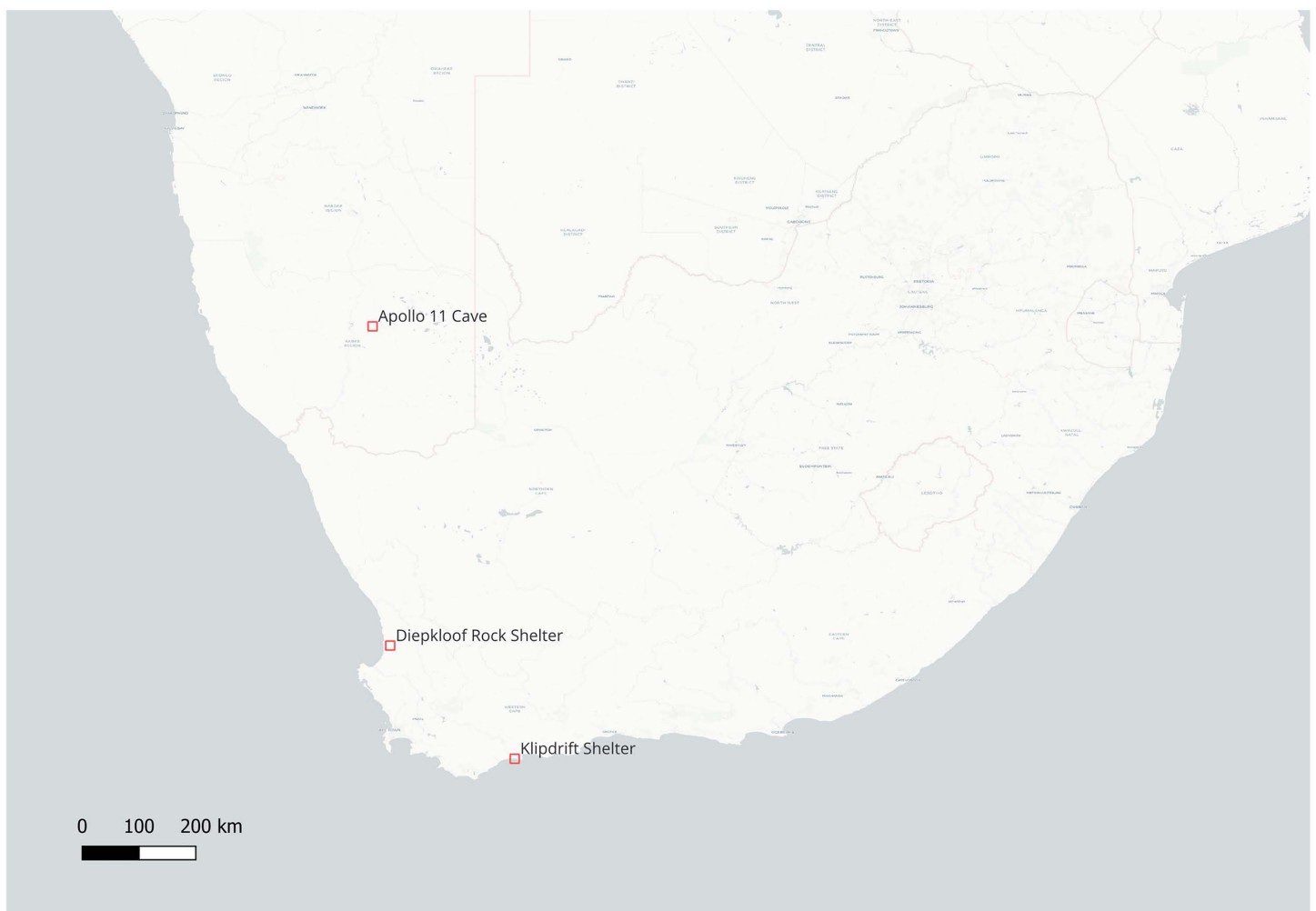

**Fig 1. Map of the HP sites with EOES.** Basemap: CARTO, © OpenStreetMap contributors, CC BY 4.0.

**Table 1. A summary of the ostrich eggshells fragments from the HP.**

| Site | Layer | EOES (nr.) | OES (nr.) | % EOES |
|------|-------|-----------|-----------|--------|
| Diepkloof Rockshelter (from 65.9±3.0 to 60.8±2.6) | Early HP | 4 | 1359 | 0.3 |
| | MSA-Jack | 4 | > 379 | < 1 |
| | Intermediate HP | 129 | > 4767 | < 2.7 |
| | Late HP | 112 | 10539 | 1 |
| | **nr. total EOES** | **249** | **> 17044** | **< 1.4** |
| Klipdrift Shelter (from 65.5±4.8 to 59.4±4.6) | PAY | 5 | 106 | 4.7 |
| | PAZ | 15 | 187 | 8.0 |
| | PBA/PBB | 22 | 1274 | 1.7 |
| | PBC | 23 | 349 | 6.6 |
| | PBD | 25 | 202 | 12.4 |
| | PBE | 4 | 90 | 4.4 |
| | PCA | 1 | 282 | 0.4 |
| | **nr. total EOES** | **95** | **2490** | **3.8** |
| Apollo 11 (63±2) | HP | 2 | 4081 | 0.05 |
| | **nr. total EOES** | **2** | **4081** | **0.05** |

Data elaborated after Texier et al. [2] for Diepkloof Rock Shelter, Henshilwood et al. [3] for Klipdrift Shelter, Ossendorf & Vogelsang [4] and Vogelsang et al. [31] for Apollo 11. The dates refer to EOES findings and for Diepkloof and Apollo 11 they follow Jacobs et al. [29].

## The dataset

In this study, the dataset of the EOES is garnered from published materials [1–5] which provided only a selection of the fragments. We collected images for 112 EOES fragments from Diepkloof (93 made available out of 249), Klipdrift (17 out of 95), and Apollo 11 (2 out of 2) (S1 Table). Twelve are joint fragments composed of 2–11 specimens. Three fragments from Diepkfloof Rockshelter (D13, D68, and D73 in our dataset) were excluded from our statistical analysis, as they appear too worn out, and the image quality hinders a precise drawing of the motifs.

## Retracing and segmenting the motifs

We manually traced the EOES motifs line by line using QGIS 3 (Fig 3), to extract their geometric and spatial features (S1 Table). We mitigated distortions issues, which are inevitable when working with second-hand material such as photographs, by visually normalizing the lines following their 'non-accidental properties' (NAPs) which are stable when viewpoint changes [32–35]. These can include curvature (a line is straight or curved), collinearity (edges align to form a continuous line), symmetry (consecutive or mirror balance), parallelism (edges remain equidistant), and cotermination (edges meet at a single point). In line with this, for example, only lines showing a clear change in direction along their path were traced as non-straight. Metric properties, as opposed to NAPs, vary continuously according to different viewpoints and are more susceptible to perceptual error (e.g., distance misestimation, angular openings [33,34]). Therefore, when classifying features, such as parallelism derived from line inclinations or right/non-right distinction derived from angular openings, we applied tolerance thresholds informed by empirical studies [35].

We define and classify geometric features and measurements as follows:

1. *Line type*: straight *vs.* non-straight. A line is defined as straight if it connects two endpoints without deviation. A non-straight line includes at least one directional change, marked by a vertex joining two segments.

2. *Parallelism*: this is assessed at the segment level. Two segments are considered parallel if their inclinations differ by no more than ±3.5°. This threshold is supported by evidence that people can discriminate between angle differences as small as 7° with approximately 53% accuracy [35].

| ID | Motif | Site & approx. dating range | | |
|----|-------|---------------------------|---|---|
| | | **Diepkloof** | **Klipdrift** | **Apollo 11** |
| 1 | Hatched-band motif (ortogonal) | ca. 63-60 ka | — | — |
| 2 | Hatched-band motif (oblique) | ca. 63-60 ka | ca. 65 - 60 ka | — |
| 3 | Crosshatched grid motif | ca. 63-60 ka | ca. 65 - 60 ka | — |
| 4 | Sub-parallel intersecting lines motif | from ca. 65 ka onwards | — | ca. 63 ka |
| 5 | Curved and sub-parallel lines motif | ca. 60 ka | ca. 65 - 60 ka | — |
| 6 | Reversed curvature motif | ca. 60 ka | ca. 65 - 60 ka | — |
| 7 | Sub-parallel rectilinear or curved lines | from ca. 65 ka onwards | ca. 65 - 60 ka | — |
| 8 | Isolated or irregular striations | from ca. 65 ka onwards | ca. 65 - 60 ka | — |
| 9 | Crosshatched diamond shaped | — | ca. 60 ka | — |

**Fig 2. A synthesis of a spatial and chronological assessment of the EOES motifs.** Data elaborated from Texier et al. [2]; Heshilwood et al. [3]; Jacobs et al. [29]. We have associated the two fragments from Apollo 11 to the 'sub-parallel intersecting lines motif' as, in our statistical analyses, they display the same values in terms of regularity of line alignments, consistent with the fragments in the same category from Diepkloof.

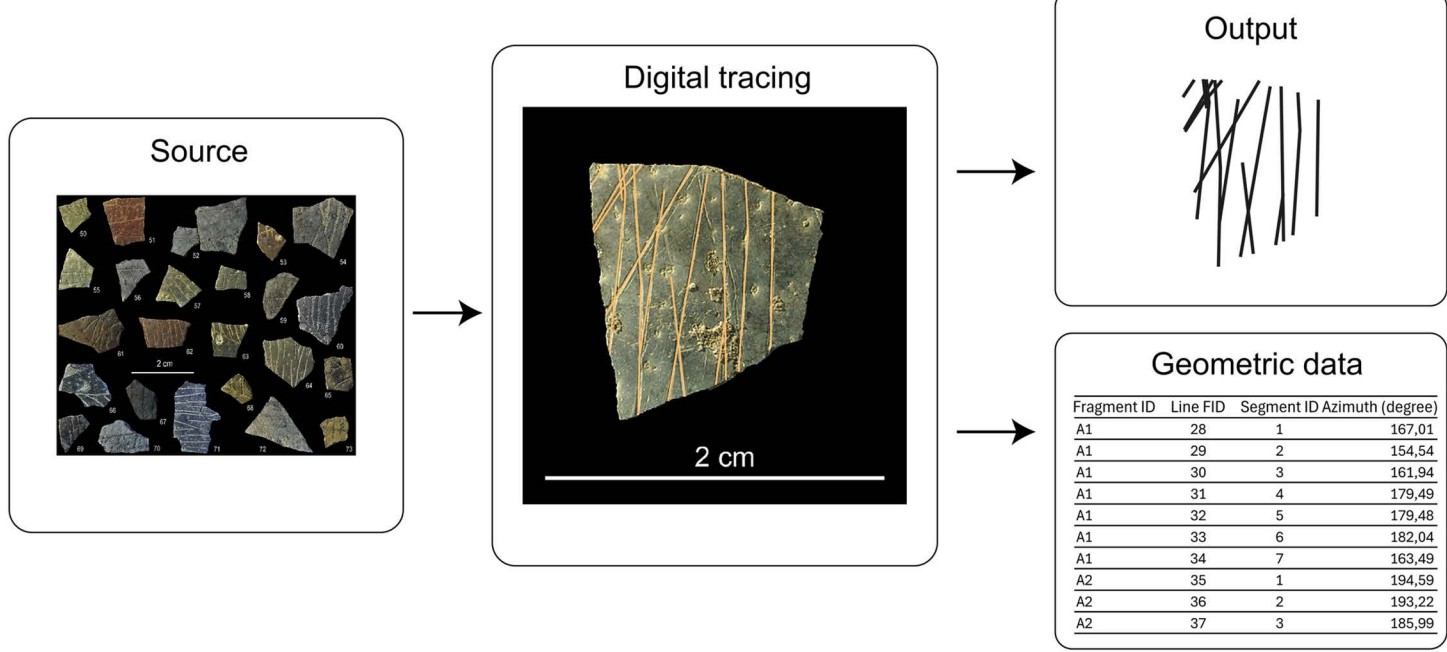

**Fig 3. Example of tracing of a fragment (modified from Texier et al. [2], Figs 8 and 3b), normalization of the engraved lines, and data extraction.**

3. *Intersection geometry*: this is detected between crossing lines, and the minor angle (the smallest angle at the intersection) is extracted for angular categorization.

4. *Angular opening*: minor angles are classified as right *vs.* non-right, with a tolerance of 7° off 90° for defining right angles.

5. *Spatial organization*: distances between intersection points are calculated from coordinates to highlight possible spatial regularities within each fragment.

These figures were then exported for statistical and comparative analysis.

## Statistical analysis

To assess whether salient features and structural regularities can be systematically recognized in the EOES motifs, we conducted statistical analyses using spreadsheets and R (detailed methods and results in S2 Document). Each analysis was specifically designed to test a distinct aspect of their structure: distributional tendencies of geometric primitives, compositional regularity in line alignment and angular relationship, patterns of spatial organization, and the salience of basic features. Taken together, these analyses offer a multi-level view of the geometric affordances at play and allow us to investigate the cognitive operations underlying motif configurations.

First, we ran a distributional analysis of the geometric features, to capture macro-level regularities in terms of salience. Specifically, we examined the frequencies and proportions of line types, parallelism, number of intersections per fragment, and distribution of minor angles (S2 Document).

Second, we determined the structural composition and complexities of the engravings. After grouping parallel segments with similar inclinations and angles with similar degrees, we applied multiple regression with residual analysis

(S2 Document), one for parallelism and one for intersection angles, examining how maximum angular spread in groups was predicted by internal structural features. This analysis allows us to statistically explain structural regularities and to identify outliers within the dataset [36,37].

Third, we tested structural regularities of intersections and angles: for example, a regular grid typically exhibits spatially ordered intersection points with consistent right angles. Therefore, we applied Moran's I to quantify the degree to which intersection angles are spatially clustered, evaluating whether angles of similar degrees are more likely to occur near each other or are randomly distributed (S2 Document). We also reported statistically non-significant cases ($p > 0.05$), as their deviations may reflect meaningful structural or compositional variation.

Finally, we applied the Principal Component Analysis (PCA) to rank the salience of these features in defining structural patterns in the dataset (S2 Document). We tested several PCA models using different variable associations, finding that the most informative results were achieved when the model incorporated the proportions of specific properties per fragment (parallel segments per total segments, intersections per total segments, right angles per total intersections), along with the measure of spatial distribution (Moran's I).

## Results and discussion

### Distribution of geometric features: the salience of parallelism and orthogonality

The dataset, derived from 109 EOES fragments, includes a total of 1275 lines (made up of 1635 individual segments) and 1405 intersections from which minor angles (1°–90°) were extracted. Results of our statistical analysis on distribution of engraved properties (S2 Document) show that the dataset is mostly composed of straight lines (78.9%) and most segments maintain a consistent inclination (83.4% of the segments were identified as belonging to parallel pairs). Most fragments (68%) contain fewer than 10 intersections, whereas a few (n = 4), particularly larger pieces or those with grid motif incisions, stand out as outliers, with over 90 intersections. A substantial proportion of the minor angles formed at these intersections (33.6%) cluster near 90°, indicating a frequent recurrence of orthogonal relationships. The regular and repeated use of these properties suggests that the OES were engraved with perceptually salient geometric affordances, namely straight/curved lines, parallel/secant segments, and right/non-right angles.

The PCA (S2 Document) further highlights orthogonality and parallelism as the main contributors at the macro level. The biplot below (Fig 4) reveals one prominent cluster (yellow), characterized by the absence of right-angles but encompassing motifs with varying degrees of parallelism and spatial organization, ranging from low regularity (bottom-left) to high alignment and consistent angular repetition (top-left). This is crucial as right angles, despite accounting for only 33.6% of the minor angles overall, emerge as the single most influential variable in the PCA, suggesting orthogonality as a key structural principle in the engravings.

A secondary, less pronounced grouping (Fig 4, red) includes fragments with both high orthogonality (presenting multiple right angles per intersections) and strong spatial regularity (parallelism and high Moran's I). In this group, parallelism gives nuance and contributes to characterizing the cluster, indicating that directional control and regularity of line placement (whether through systematic intersection or alignment) play a foundational role in the construction of these patterns. Together, orthogonality (most influential component, Dim1) and line alignment and spatial regularity together (correlated variables, Dim2) feature in over 80% of the dataset. This is in line with previous research on the cognitive capacity to visually discriminate shapes that present these geometric primitives [35]. The remaining fragments are more diffusely distributed and represent hybrid or less systematically structured compositions.

It is important to note that the PCA does not reveal clear-cut or sharply separated motif types. Instead, it brings to light the most representative features used by the engravers to construct the motifs. As described in the following sections, these features are attested in a systematic way through recurrent cognitive operations (iteration, rotation, translation, and embedding), mechanisms which are all at play to produce visually organized and structured configurations.

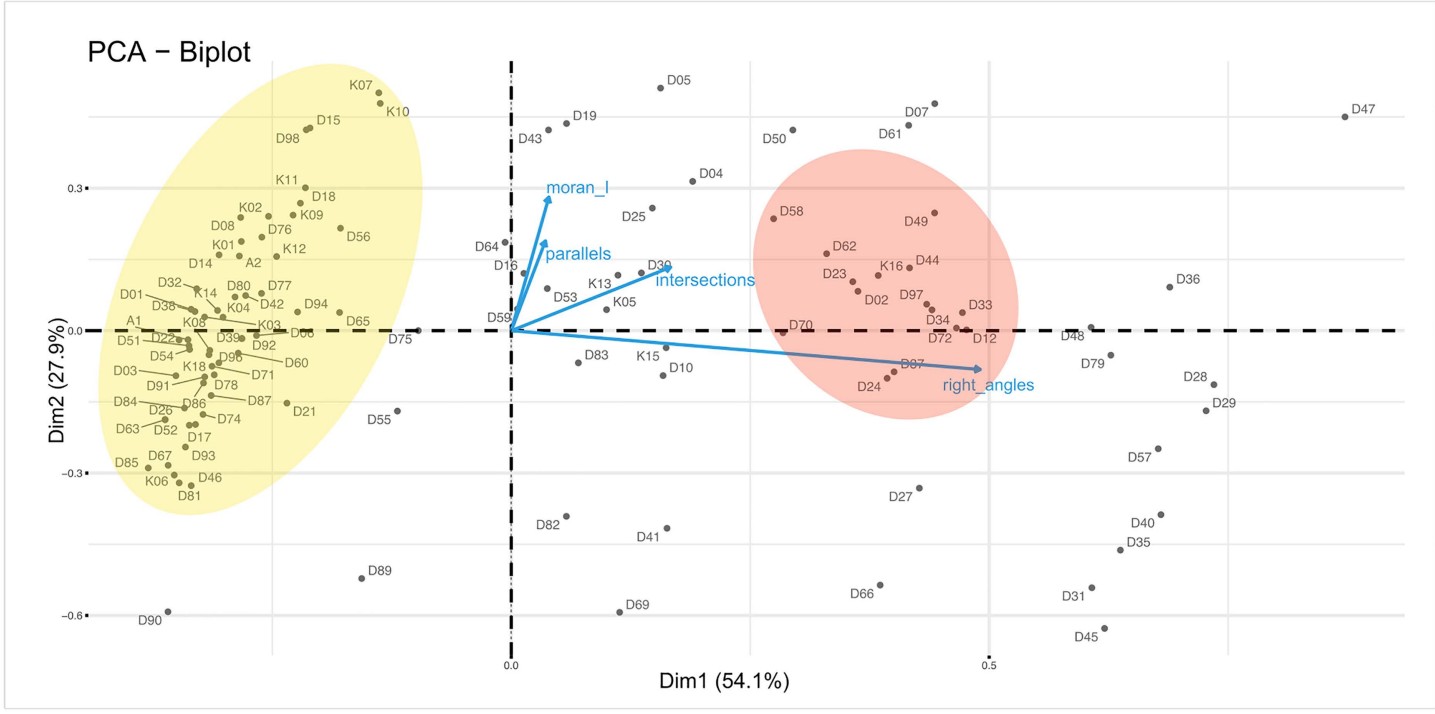

**Fig 4. PCA biplot of the EOES fragments showing clusters of geometric patterns highlighted in yellow and red.**

### Regularities in alignment strategies and angular patterns: Toward a 'geometric grammar'

Each eggshell engraving was analyzed in relation to its internal organization, focusing on both structural composition and level of complexity. Regression analysis on parallelism (S2 Document) shows that both the total number of segments and the number of parallel groups significantly predicted angular spread, yielding a strong model fit ($R^2 = 0.736$). In contrast, for minor angles, only the number of intersections emerged as a significant predictor, while grouping did not reach significance, likely due to both limited variance and the presence of 17 fragments with no angular grouping. Residual analysis confirms that most fragments closely conform to the predictions of the model: 91.6% for parallelism (out of 107 fragments) and 93.0% for minor angles (out of 86 fragments). This indicates that, for most of the dataset (>90%), alignment in parallelism and consistent angular opening manifest a high level of structural regularity, involving visual planning and compositional intent.

However, a small set of outliers deviates from model expectations and offers particularly informative cases. Regarding parallelism, nine outliers were identified, six with positive residuals, indicating greater angular spread than expected, and three with negative residuals. Fragments with positive residuals, such as D14, D52, and D58 (Fig 5), likely reflect looser alignment strategies, imprecise execution (motor or tool constraint), responses to material factors such as curvature or irregularities on the eggshell surface, or compositional layering. Therefore, these outliers indicate that statistical irregularity is not necessarily correlated to structural arrangement. Specifically, D58 seems to be a case of compositional layering as we can distinguish a conjunction of two grids with slightly different inclinations (S1 Table).

This result could refine some of the previous analyses [2] and may suggest that some eggs displayed multiple patterns – as shown also in the record of extant foragers [38,39]. Conversely, negative residuals such as D49, D50, and D98 (Fig 6) show far lower spread than predicted, pointing to exceptional control and precision, deliberate alignment, and strong adherence to spatial organization. While these pieces present visual irregularities, their large size makes

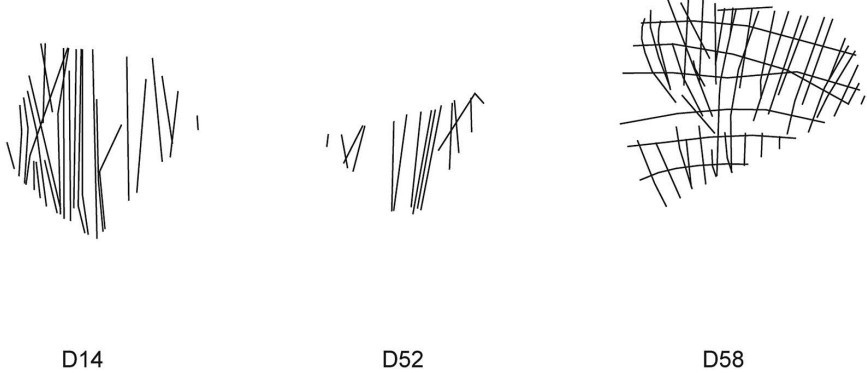

D14          D52          D58

**Fig 5. Positive outliers in the multiple regression analysis.** These cases show a low degree of alignment strategies.

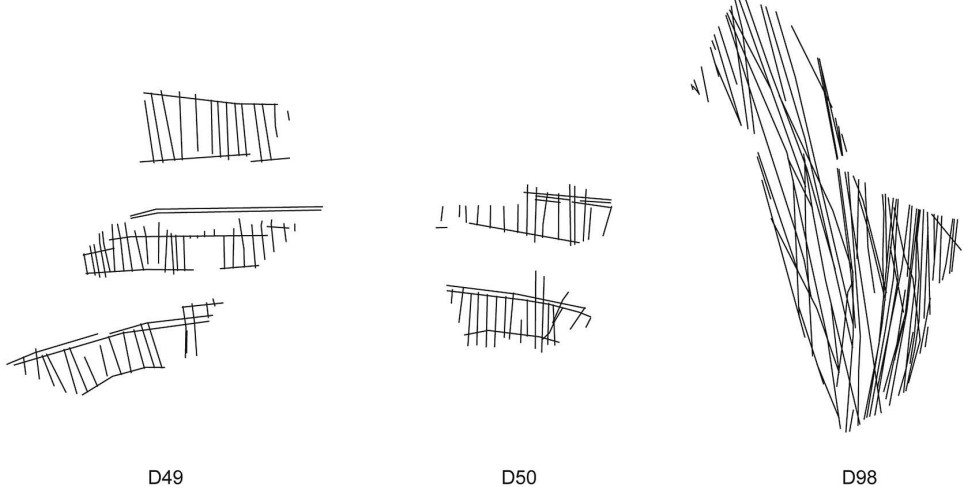

D49          D50          D98

**Fig 6. Negative outliers in the multiple regression analysis.** The engravings exhibit an extremely high level of regularity in terms of parallelism. D49 shows significantly high structural consistency also in terms of minor angle degree.

these inconsistencies statistically irrelevant in terms of the overall structure. These cases may reflect a stronger cognitive emphasis on regularity as a visually and conceptually salient feature, rather than the product of mechanical repetition.

A comparable trend emerges in the minor angle model, which identifies six outliers. Positive residuals – including D04, and D10 – indicate higher-than-expected angular variability, possibly due to overlapping gestures or surface-induced distortions. As with parallelism, these deviations show that angular irregularity is not solely a byproduct of compositional density of intersections. The single negative outlier, D49 (Fig 6), again shows striking angular precision, reinforcing its interpretation as a particularly deliberate and controlled composition, sensitive to both angular regularity and parallel alignment. Taken together, these results show that over 90% of the EOES fragments exhibit structural and consistent geometric organization, with angular variation well explained by the internal complexity and the structure of each engraving.

This structural coherence of the dataset results from a series of operations applied to sequences of aligned parallel lines and repeated angular configurations, such as rotation (moving a feature around a fixed point by a certain number of degrees), translation (moving a feature to a given point in space maintaining its size and inclination), iteration (repeating

features within the same hierarchical level), and embedding (nested features in additional hierarchical levels) (Fig 7). The engravings are created not just through the repetition of these features or operations, but through ruled-based procedures that can become superimposed, governing how features are related and assembled into complex patterns. For example, a set of iterated parallel lines can provide embedding for other sets of lines. Thus, combinatorial patterns can create an internal logic that constitutes a foundational 'geometric grammar'.

This internal geometric logic becomes especially evident in the hatched-band, grid, and diamond-shaped motifs, which show a deliberate construction of large, structurally coherent configurations through precise alignment and adherence to spatial organization. Hatched-band motifs, for instance, are created by first drawing two parallel lines that define a space within which another line, starting from the upper parallel and ending at the lower one, is iterated (Fig 7−1). This line, inclined up to 90° relative to the two parallel baselines, is then translated *n* times. The embedding of this second set of lines within the first results in hierarchically organized configurations. Interestingly, the different observed inclinations (acute *vs.* right angle) may imply distinct perceptual processing, as shape discrimination changes when rotation exceeds 40° [40].

Grid motifs show comparable operations: parallel segments are iterated to define a space within which other parallel segments, inclined 90° and starting from the upper parallel line, are embedded (Fig 7−2), as suggested by the upper-left portion of D19 in our dataset (S1 Table). Diamond-shaped motifs, by contrast, stand out because the space in which the configuration is embedded is not outlined by visible lines, but rather emerges as an abstract area made perceptible through the regularity of the composition itself, where iterative operations unfold (Fig 7−3; K1, K2, K7, K10, K11 in S1 Table).

Crucially, these findings resonate with experimental studies on both geometrically literate and illiterate adults and children, which demonstrate a shared cognitive capacity for combining and embedding shapes according to operation-based structures [15,18,19]. The essential characteristics of the EOES motifs are rooted in these shared cognitive capabilities and reflect a visuo-spatially grounded and planned mode of composition.

## Spatial arrangements of intersection geometries: regular distances

Regularities in geometric structures of the EOES motifs are evident not only in alignment and angular consistency, but also in the spatial distribution of line intersection points and their associated angular values. Moran's I was performed on 76 fragments as 27 were excluded due to an insufficient number of intersections (n < 4). The results (S2 Document) show that 29 fragments have positive Moran's I, indicating that similar angular values tend to be displayed at a regular distance (Fig 8). This reinforces that maintaining consistent spacing is an integral part of the procedure for constructing the motifs, as observable in the iterative process of translation at a fixed distance (Fig 7). In contrast, 47 fragments yield negative Moran's I, suggesting that the intersection points tend to be spatially distributed but display varying angle values, therefore they lack regularity in alignment. This is consistent with the results of the regression analyses, which revealed higher-than-expected angular variability, explained by the overlap of layers or surface-induced distortions.

Interestingly, there are 11 fragments that present a positive Moran's I but with no significant *p-value* (p > 0.05). These cases present spatial regularities in terms of intersection points, but lack consistency due to the low number of intersections or noise in the spatial arrangement (Fig 9). In the first case, the fragmented pattern in these engravings has evidently influenced the results, as the missing parts alter the spatial distribution under examination. The second case presents suggestive variations that diverge from regular patterns. For example, in some fragments, the hatched-band motifs consist of regularly spaced parallel lines that do not intersect with the upper or lower band (e.g., D36 and D37 in Fig 9). In other fragments, the horizontal top/bottom lines of the hatched band are not aligned (e.g., D02). Some others include a series of parallel lines intercepted by oblique lines (e.g., D04 and K05). Lastly, some fragments present more than one of the above variations (e.g., D02, D12, D65, D70).

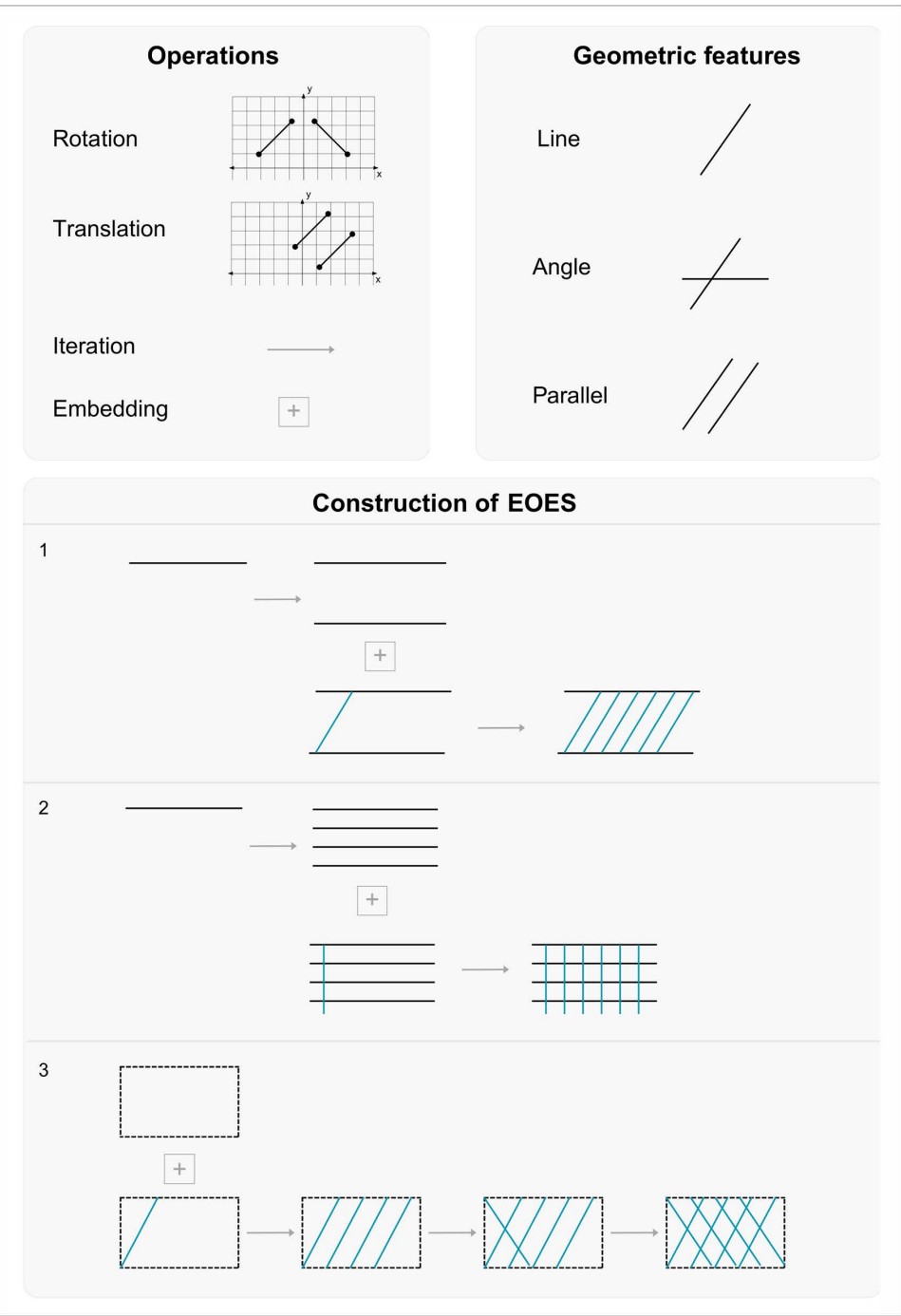

**Fig 7. Repertoire of the operations underlying the 'geometric grammar' in the EOES.** These operations illustrate the geometric primitives and the embedding processes identifiable in the EOES motifs (1: hatched band; 2: grid; 3. diamond shape). This image was the result of a pseudo-code (plain language algorithm) in S2 Document.

MOTIFS with positive Moran's I          MOTIFS with negative Moran's I

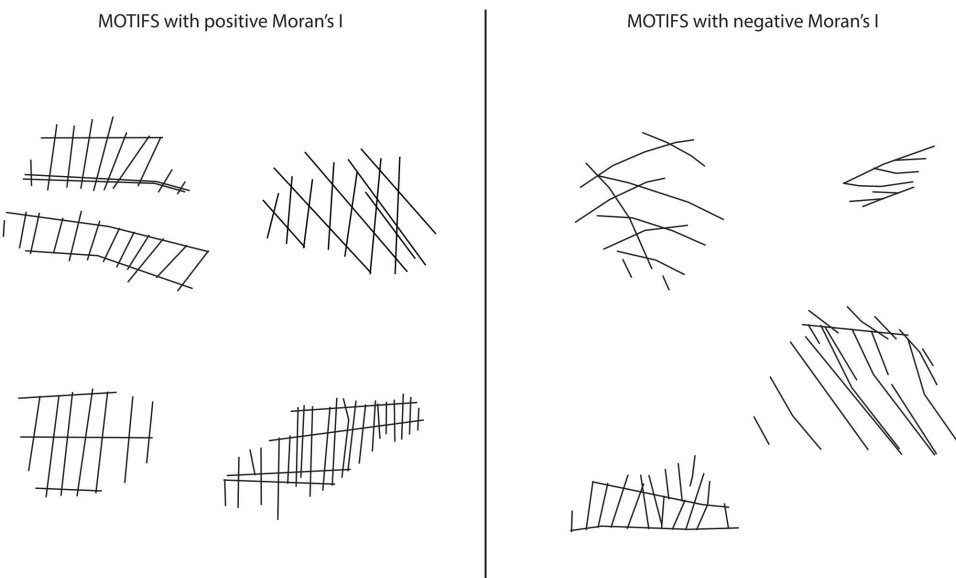

**Fig 8. Examples of EOES motifs with positive and negative Moran's I.**

*Variation 1*
Orthogonal lines
not intersecting
with horizontal band

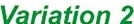

*Variation 2*
Hatched-band with
straight and oblique lines

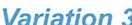

*Variation 3*
Hatched-band
with not aligned
horizontal bands

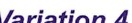

*Variation 4*
Two parallel long lines
in hatched-band motif

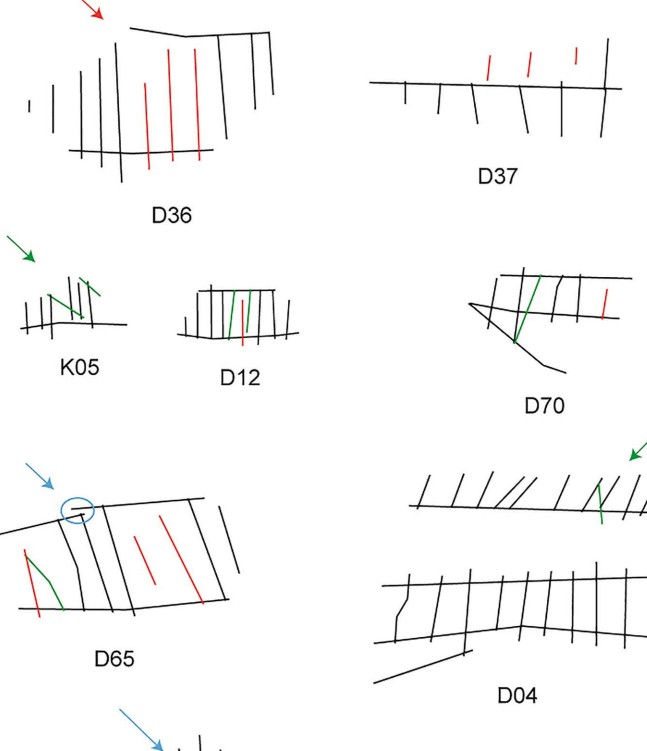

**Fig 9. Variations in motif composition on the EOES.**

These departures are perceptually salient as they deviate from the expected recurrent spatial consistency [41,42]. The high degree of regularity observed in our analyses indicates technical skills with a precise execution of structured patterns (see for example larger rejoined fragments such as D49, D50, and D98, Fig 6), which suggests intentional variation. To further verify this possibility, specific analyses supported by experimental work and hands-on examination of the material are required and deserve a separate investigation.

## Conclusion

This analysis provides the first method-based quantitative assessment of the geometric arrangements of the EOES within the HP technocomplex. The results demonstrate that *Homo sapiens* during the late MSA mastered precise, pre-planned patterns anchored in specific geometric affordances: orthogonality and parallelism. These affordances were related and organized through a distinct set of cognitive operations: rotation, translation, iteration, and embedding.

Our dataset is particularly well represented by rotation, translation, and iteration, as these operations enable the construction of simple yet recurrent combinations, including aligned sets of lines and repeated angular configurations. Notably, the analysis revealed that some of these combinations generate a defined spatial framework within which embedding occurs. This framework represents a zero level—a reference plane—within which features and operations are nested to form highly structured, regularly spaced patterns such as hatched bands, grids, and diamonds.

What emerges is a 'geometric grammar', that is a coherent and procedural system of operations through which patterns are constructed according to a stable internal logic. Consequently, the EOES embody the cognitive potential and foundational principles underlying symbol making, that is the ability to iterate features, embed structures hierarchically, and maintain visual coherence through abstraction. These engravings therefore constitute an early material expression of complex graphic representation, attesting to a species-specific human capacity for organizing geometric thought.

## Supporting information

**S1 Table. EOES dataset.**
(PDF)

**S2 Document. Methods and results.**
(PDF)

## Author contributions

**Conceptualization:** Silvia Ferrara, Enza Elena Spinapolice.

**Data curation:** Valentina Decembrini, Ludovica Ottaviano, Mattia Cartolano.

**Formal analysis:** Valentina Decembrini, Ludovica Ottaviano, Mattia Cartolano.

**Funding acquisition:** Silvia Ferrara.

**Investigation:** Valentina Decembrini, Ludovica Ottaviano.

**Methodology:** Valentina Decembrini, Ludovica Ottaviano, Mattia Cartolano, Enza Elena Spinapolice, Silvia Ferrara.

**Project administration:** Silvia Ferrara.

**Resources:** Valentina Decembrini, Mattia Cartolano.

**Software:** Valentina Decembrini.

**Supervision:** Silvia Ferrara, Ludovica Ottaviano, Enza Elena Spinapolice.

**Validation:** Valentina Decembrini, Ludovica Ottaviano, Enza Elena Spinapolice, Silvia Ferrara.

**Visualization:** Ludovica Ottaviano.

**Writing – original draft:** Valentina Decembrini, Ludovica Ottaviano, Mattia Cartolano.

**Writing – review & editing:** Valentina Decembrini, Ludovica Ottaviano, Silvia Ferrara.

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
