## [Decision Letter · Decision Letter 0]

22 Sep 2025

Dear Dr. Ferrara,

Thank you for submitting your manuscript to PLOS ONE. After careful consideration, we feel that it has merit but does not fully meet PLOS ONE’s publication criteria as it currently stands. Therefore, we invite you to submit a revised version of the manuscript that addresses the points raised during the review process.

We look forward to receiving your revised manuscript.

Kind regards,

Iris Groman-Yaroslavski, Ph.D

Academic Editor

PLOS ONE

2. In your manuscript, please provide additional information regarding the specimens used in your study. Ensure that you have reported human remain specimen numbers and complete repository information, including museum name and geographic location.

For more information on PLOS One's requirements for paleontology and archeology research, see https://journals.plos.org/plosone/s/submission-guidelines#loc-paleontology-and-archaeology-research .

Additional Editor Comments (if provided):

Reviewer #1:

Reviewer #2:

Reviewers' comments:

Reviewer's Responses to Questions

**Comments to the Author**

1. Is the manuscript technically sound, and do the data support the conclusions?

Reviewer #1: Yes

Reviewer #2: Partly

2. Has the statistical analysis been performed appropriately and rigorously?

Reviewer #1: I Don't Know

Reviewer #2: Yes

3. Have the authors made all data underlying the findings in their manuscript fully available?

Reviewer #1: Yes

Reviewer #2: Yes

4. Is the manuscript presented in an intelligible fashion and written in standard English?

Reviewer #1: Yes

Reviewer #2: Yes

Reviewer #1: Reviewer recommendation for manuscript number PONE-D-25-36789: “Earliest geometries. A cognitive investigation of Howiesons Poort engraved ostrich eggshells”

General remarks – the manuscript presents in intriguing idea, and interpretive framework for the early geometric patterns of EOES. Yet the large theoretical background described in the abstract and introduction do not seem to be validated, at least not as described in the conclusions. It is possibly not well explained, but the conclusions seem to be more “down to earth” compared to what is presented in the introduction. Thus, there seems to be a disconnect between the introduction and the discussion, which should be readjusted to focus more on the cognitive operations, and the transmission and transformation of the geometric forms.

In general, there is tendency for over-complication in terms of language, I would recommend careful read through of the manuscript to simplify words and restructure complicated sentences.

Overall, this could be a very interesting paper, after major revisions, simplifying the language and ideas, clarifying the methodology, and tying the theoretical background to the practical work.

Abstract

• The abstract is too general and theoretical and doesn’t really describe the essentials of the research. It should include the research question, the analytical tools used to tackle the question, as well a succinct description of the results and conclusions.

Method –

• There are several problems in the methodology, or points that could use clarification. It is understood that the data derives from published photographs, but this means that the authors have no control over positioning of the fragments. A non-uniform way of positioning will alter the lines, spatial relations between lines (parallelness etc.), the measurement of the angles, curvature, as well as distances between points. The authors acknowledge only the possible distortion in discussing metric measurements, but do not explicitly explain why they are measured anyway. If the measurements are skewed, can the information collected from them be crucial? The bias should be acknowledged for all parameters.

• What does tracing “both manually and digitally” mean? (Line 122) It seems like manual tracing in a digital format (QGIS)

• “Visual normalization” is unclear, does this mean that the normalization is particular to each etched line, as opposed to normalizing the assemblage? Line 125 - Citation of Biederman 1987 – a short sentence explaining the idea and the relevance of the theory to the present research. It seems a crucial reference to understand the method but a reference alone is not enough.

• In the methods section there is a list of numbered points (from line 129), which are not entirely clear. Do they refer to the tolerance thresholds? Or are they steps in the methodology? If so, it should be differentiated in the text as a sub-title within the methodology. The points could include references to figures or possibly replace by a figure illustrating and outlining the analytical steps; alongside an illustration of the differentiation between lines and segments (and their “directionality”) and examples of why and how they are segmented.

Reviewer #2: This paper addresses an interesting question in the field of cognitive archaeology, that of what cognitive mechanisms underlie graphic symbolism in our species. The authors attempt to contribute to answering that question by analysing a corpus of early visual sign manifestations: the engraved ostrich eggshells (OES) found at the South African sites of Diepkloof and Klipdrft Shelter, and Apollo 11 in Namibia, all dating back some 60-65,000 years to an archaeological complex known as Howiesons Poort.

The present study carries out the first quantitative analyses of the OES with the aim to identify the cognitive processes that supported the creation of the engravings. Whereas the methods seem meticulous and raise important questions about the materials and their creators, I also found several shortcomings, detailed below.

1. General.

One of the main claims of the study seems to be that the engraved OES show intentionality, awareness, and precise execution. This is backed by the precise quantitative analyses presented by the authors. However, the intentionality of the engravings has never been in question. Since they were first published, several papers have argued that the motifs were made purposefully and originally constituted complex geometric patterns (e.g., Henshilwood et al., 2014; Nel and Haaland, 2023; Texier et al., 2010; 2013; Tylén et al., 2020 Vogelsang et al., 2010). Also, the earlies Blombos engraving has already been identified as following geometric primitives. In this sense, I wonder what this paper contributes to the already ongoing discussion about early symbolism, other than another method of analysing the geometric patterns.

2. Introduction

The introduction goes into an interesting discussion on the cognitive fundaments of geometry and spatial cognition, but I feel this is not followed up in the discussion with sufficient depth. After having shown that the OES patterns were conceived following geometric ‘rules’, it would have been interesting to read why this is relevant to human cognition during the MSA.

3. Materials and Methods

a) Regarding the materials and methods, it is understandable that the authors worked from published materials, and this practice is common in archaeology. However, I miss a limitations section where it is explicitly acknowledged that working from second hand sources has constraints, along with issues of working with fragmentary material, and the chronological span between samples, which I feel is much underplayed throughout the paper (despite all materials belonging to the HP, there are potentially five millennia between some of them).

b) The dataset section should include the sample size with exclusions (these are mentioned in the results section but should be described in the materials and methods section).

4. Results and discussion.

a) In my opinion, there are several statements that should be elaborated. For example, line 205 reads: “EOES were engraved with perceptually grounded basic geometric affordances, that is straight/curved lines, parallel/ secant segments, and right/non-right angles, and cognitively salient strategies”. I don’t fully grasp what is meant with ‘cognitively salient strategies’. Are the authors referring to visual saliency of low-level features, or to processing fluency? In general, the terminology should be better explained for readers who are not too familiar with cognitive science lingo.

b) Line 291 claims that the patters exhibit a “cognitively traceable ‘grammar’”. This is an interesting conclusion but sadly it leads nowhere. For this to be supported, the authors should demonstrate if and how the production process follows a recurrent series of steps across the sample (e.g., certain types of lines often follow from others in a sequence), as it has been shown to be the case for stone-tool technologies (Stout et al., 2021). The mere repetition of segments and angles may show intentionality but does not constitute an action grammar. If I misunderstood, and this is actually what they found, it should be explained better and discussed in the framework of relevant literature.

c) I miss a discussion on the internal variability of the sample. I mean, it would have been interesting to see whether and to what extent the fragments of each site differed from each other, perhaps alluding to different production strategies or local pattern preferences.

d) Line 349 starts discussing pattern variations and their possible causes (different tools, skill levels, etc.). Again, this is an interesting discussion that falls short. Elaborating how different tools and skills could have led to different patterns (supported ideally by experimental work) would offer a novel approach to these materials, in the way this has been done for figurative engravings (Rivero, 2016).

5. Conclusion

The conclusion that this study offers a new way of understanding “the mechanisms by which geometric strategies were organized, transmitted, transformed, and stabilized within early human communities” (Line 369) seems to me an overstatement, given the gaps mentioned above. The following lines suggest that some of the questions raised in this review might be addressed in future work, so perhaps this analysis was premature and the authors should have waited until they could offer an more robust paper.

Cited references:

Henshilwood, C. S., van Niekerk, K. L., Wurz, S., Delagnes, A., Armitage, S. J., Rifkin, R. F., ... & Mienies, S. S. (2014). Klipdrift shelter, southern Cape, South Africa: preliminary report on the Howiesons Poort layers. Journal of Archaeological Science, 45, 284-303.

Nel, T. H., & Haaland, M. M. (2023). Klipdrift Shelter, South Africa. In Handbook of Pleistocene Archaeology of Africa: Hominin behavior, geography, and chronology (pp. 1549-1561). Cham: Springer International Publishing.

Rivero, O. (2016). Master and apprentice: evidence for learning in Palaeolithic portable art. Journal of Archaeological Science, 75, 89-100.

Stout, D., & Chaminade, T. (2012). Stone tools, language and the brain in human evolution. Philosophical Transactions of the Royal Society B: Biological Sciences, 367(1585), 75-87.

Texier, P. J., Porraz, G., Parkington, J., Rigaud, J. P., Poggenpoel, C., & Tribolo, C. (2013). The context, form and significance of the MSA engraved ostrich eggshell collection from Diepkloof Rock Shelter, Western Cape, South Africa. Journal of Archaeological Science, 40(9), 3412-3431.

Texier, P. J., Porraz, G., Parkington, J., Rigaud, J. P., Poggenpoel, C., Miller, C., ... & Verna, C. (2010). A Howiesons Poort tradition of engraving ostrich eggshell containers dated to 60,000 years ago at Diepkloof Rock Shelter, South Africa. Proceedings of the National Academy of Sciences, 107(14), 6180-6185.

Tylén, K., Fusaroli, R., Rojo, S., Heimann, K., Fay, N., Johannsen, N. N., ... & Lombard, M. (2020). The evolution of early symbolic behavior in Homo sapiens. Proceedings of the National Academy of Sciences, 117(9), 4578-4584.

Vogelsang, R., Richter, J., Jacobs, Z., Eichhorn, B., Linseele, V., & Roberts, R. G. (2010). New excavations of Middle Stone Age deposits at Apollo 11 Rockshelter, Namibia: stratigraphy, archaeology, chronology and past environments. Journal of African Archaeology, 8(2), 185-218.

**Do you want your identity to be public for this peer review?** For information about this choice, including consent withdrawal, please see our Privacy Policy

Reviewer #1: No

Reviewer #2: No

---

## [Author Response · Author response to Decision Letter 1]

24 Oct 2025

PONE-D-25-36789

Earliest geometries. A cognitive investigation of Howiesons Poort engraved ostrich eggshells

PLOS ONE

Dear Iris Groman-Yaroslavski,

Dear Editors,

Thank you for your assistance and for sharing the information regarding the journal requirements and reviewer comments. Please find our responses below, addressing both the journal requirements and the reviewers’ comments.

Journal requirements

We confirm that our revised manuscript fully meets PLOS ONE’s style and file-naming requirements.

2. In your manuscript, please provide additional information regarding the specimens used in your study. Ensure that you have reported human remain specimen numbers and complete repository information, including museum name and geographic location. If permits were required, please ensure that you have provided details for all permits that were obtained, including the full name of the issuing authority, and add the following statement: 'All necessary permits were obtained for the described study, which complied with all relevant regulations.' If no permits were required, please include the following statement: 'No permits were required for the described study, which complied with all relevant regulations.' For more information on PLOS One's requirements for paleontology and archeology research, see https://journals.plos.org/plosone/s/submission-guidelines#loc-paleontology-and-archaeology-research.

We confirm that our study is based exclusively on previously published materials. No new excavation, handling, or analysis of human remains or archaeological specimens was conducted.

Thank you. We have verified the information and corrected the grant details to ensure consistency between the Funding Information and Financial Disclosure sections. The correct grant numbers are now provided in the Funding Information section.

4.1. You may seek permission from the original copyright holder of Figure 1 to publish the content specifically under the CC BY 4.0 license. We recommend that you contact the original copyright holder with the Content Permission Form (http://journals.plos.org/plosone/s/file?id=7c09/content-permission-form.pdf) and the following text: “I request permission for the open-access journal PLOS ONE to publish XXX under the Creative Commons Attribution License (CCAL) CC BY 4.0 (http://creativecommons.org/licenses/by/4.0/). Please be aware that this license allows unrestricted use and distribution, even commercially, by third parties. Please reply and provide explicit written permission to publish XXX under a CC BY license and complete the attached form.” Please upload the completed Content Permission Form or other proof of granted permissions as an "Other" file with your submission. In the figure caption of the copyrighted figure, please include the following text: “Reprinted from [ref] under a CC BY license, with permission from [name of publisher], original copyright [original copyright year].”

4.2. If you are unable to obtain permission from the original copyright holder to publish these figures under the CC BY 4.0 license or if the copyright holder’s requirements are incompatible with the CC BY 4.0 license, please either i) remove the figure or ii) supply a replacement figure that complies with the CC BY 4.0 license. Please check copyright information on all replacement figures and update the figure caption with source information. If applicable, please specify in the figure caption text when a figure is similar but not identical to the original image and is therefore for illustrative purposes only.

We have a new Figure 1 that meets PLOS ONE’s requirements. This version was created in QGIS using as basemap CARTO, © OpenStreetMap contributors, CC BY 4.0 and it will be referenced in the caption.

Also, we have revised the supplementary material 'S1 Table', removing all copyrighted content to ensure full compliance with publication and licensing requirements.

N/A.

We have uploaded all figure files (Figures 1–9) to the PACE tool, as requested, and updated them to meet PLOS requirements. Please note that:

- Figure 1 has been completely redone;

- Figure 7 is a newly added figure;

- Figure 8 (previously Figure 7) has been edited, with minor textual modifications;

- Figure 9 (previously Figure 8) has been updated accordingly.

Response to Reviewers

We thank the reviewers for their insightful comments which have given us the opportunity to significantly improve our paper. We agree with the reviewers on the necessity of revising the manuscript to better reflect the importance of our study and to highlight its cognitive perspective. We significantly reviewed the manuscript accordingly, in order to tie the sections harmoniously, through major adjustments and insertions. Our responses (in blue below) address and explain our changes, and report the specific lines in the manuscript which we have modified. In the manuscript, we have kept the revisions in track changes.

Reviewer #1

General remarks – the manuscript presents in intriguing idea, and interpretive framework for the early geometric patterns of EOES. Yet the large theoretical background described in the abstract and introduction do not seem to be validated, at least not as described in the conclusions. It is possibly not well explained, but the conclusions seem to be more “down to earth” compared to what is presented in the introduction. Thus, there seems to be a disconnect between the introduction and the discussion, which should be readjusted to focus more on the cognitive operations, and the transmission and transformation of the geometric forms.

In general, there is tendency for over-complication in terms of language, I would recommend careful read through of the manuscript to simplify words and restructure complicated sentences.

Overall, this could be a very interesting paper, after major revisions, simplifying the language and ideas, clarifying the methodology, and tying the theoretical background to the practical work.

We are grateful for this constructive comment. We have substantially rewritten the manuscript to improve clarity, simplify language, and strengthen alignment between theoretical framing and the results. The introduction, in particular, has been refocused on the cognitive operations that can be empirically reconstructed, and the discussion and conclusion now explicitly address these operations and the results that support them. We have striven to tie the sections of the manuscript together, and clarified the overall structure, with heavy interventions in the conclusions as well. Moreover, we revised the titles of the three subsections in the ‘Results and discussion’ section to better connect the statistical analyses to the cognitive implications. The new titles are as follows: ‘Distribution of geometric features: the salience of parallelism and orthogonality’ (originally ‘Distribution of geometric features and their salience in the whole dataset’), ‘Regularities in alignment strategies and angular patterns: toward a ‘geometric grammar’’ (originally ‘Regularities in alignment strategies and angular patterns’), and ‘Spatial arrangements of intersection geometries: regular distances’ (originally ‘Variations from regular patterns and their salience’). We are confident that the manuscript in this latest version reflects the results we have reached and conveys clearly our conclusions.

Abstract

• The abstract is too general and theoretical and doesn’t really describe the essentials of the research. It should include the research question, the analytical tools used to tackle the question, as well a succinct description of the results and conclusions.

Thank you, we have rewritten the abstract to describe more clearly the aims, methods, and results of our analysis.

Method

• There are several problems in the methodology, or points that could use clarification. It is understood that the data derives from published photographs, but this means that the authors have no control over positioning of the fragments. A non-uniform way of positioning will alter the lines, spatial relations between lines (parallelness etc.), the measurement of the angles, curvature, as well as distances between points. The authors acknowledge only the possible distortion in discussing metric measurements, but do not explicitly explain why they are measured anyway. If the measurements are skewed, can the information collected from them be crucial? The bias should be acknowledged for all parameters.

We now explicitly explain in the manuscript that, although possible distorsions are inevitable when dealing with photographs, we have normalized lines working with non-accidental properties (NAPs). This is crucial as NAPs do not alter with changing viewpoints. Therefore this should strengthen the results via the method we used (Please see lines 359-364 in the manuscript):

‘We mitigated distortion issues, which are inevitable when working with second-hand material such as photographs, by visually normalizing the lines following their ‘non-accidental properties’ (NAPs) which are stable when viewpoint changes [32-35]. These can include curvature (a line is straight or curved), collinearity (edges align to form a continuous line), symmetry (consecutive or mirror balance), parallelism (edges remain equidistant), and co-termination (edges meet at a single point).’

Conversely, when measurements are included, which are instead susceptible to distortions, we applied empirically defined tolerance thresholds on angular deviations (for parallelism and orthogonality), and relational distance of points rather than absolute metrics for spatial organization. This is stated in the manuscript in lines 418-422:

‘Metric properties, as opposed to NAPs, vary continuously according to different viewpoints and are more susceptible to perceptual error (e.g., distance misestimation, angular openings [33,34]). Therefore, when classifying features, such as parallelism derived from line inclinations or right/non-right distinction derived from angular openings, we applied tolerance thresholds informed by empirical studies [35].’

This ensures that deviations are not only acknowledged, but taken into account and mitigated, and that non-accidental properties will in any case preserve perspective.

• What does tracing “both manually and digitally” mean? (Line 122) It seems like manual tracing in a digital format (QGIS)

We have clarified this point as suggested. Please see lines 358-359:

‘We manually traced the EOES motifs line by line using QGIS 3 (Fig 3), to extract their geometric and spatial features’.

• “Visual normalization” is unclear, does this mean that the normalization is particular to each etched line, as opposed to normalizing the assemblage? Line 125 - Citation of Biederman 1987 – a short sentence explaining the idea and the relevance of the theory to the present research. It seems a crucial reference to understand the method but a reference alone is not enough.

We very much appreciate the suggestion. We have refined the explanation on ‘visual normalization’ since what we have specifically focused on are Non Accidental Properties of the markings and explained them as indicated in the comment above.

• In the methods section there is a list of numbered points (from line 129), which are not entirely clear. Do they refer to the tolerance thresholds? Or are they steps in the methodology? If so, it should be differentiated in the text as a sub-title within the methodology. The points could include references to figures or possibly replace by a figure illustrating and outlining the analytical steps; alongside an illustration of the differentiation between lines and segments (and their “directionality”) and examples of why and how they are segmented.

We have implemented a series of changes throughout this subsection to clarify the definitions and methodological principles of our retracing of the EOES (the numbered points). Please see specifically lines 423-436:

‘We define and classify geometric features and measurements as follows:

1. Line type: straight vs. non-straight. A line is defined as straight if it connects two endpoints without deviation. A non-straight line includes at least one directional change, marked by a vertex joining two segments.

2. Parallelism: this is assessed at the segment level. Two segments are considered parallel if their inclinations differ by no more than ±3.5°. This threshold is supported by evidence that people can discriminate between angle differences as small as 7° with approximately 53% accuracy [35].

3. Intersection geometry: this is detected between crossing lines, and the minor angle (the smallest angle at the intersection) is extracted for angular categorization.

4. Angular opening: minor angles are classified as right vs. non-right, with a tolerance of 7° off 90° for defining right angles.

5. Spatial organization: distances between intersection points are calculated from coordinates to highlight possible spatial regularities w

---

## [Decision Letter · Decision Letter 1]

25 Nov 2025

Earliest geometries. A cognitive investigation of Howiesons Poort engraved ostrich eggshells

PONE-D-25-36789R1

Dear Dr. Ferrara,

We’re pleased to inform you that your manuscript has been judged scientifically suitable for publication and will be formally accepted for publication once it meets all outstanding technical requirements.

Kind regards,

Iris Groman-Yaroslavski, Ph.D

Academic Editor

PLOS ONE

Additional Editor Comments (optional):

Reviewers' comments:

Reviewer's Responses to Questions

**Comments to the Author**

Reviewer #1: All comments have been addressed

Reviewer #2: All comments have been addressed

2. Is the manuscript technically sound, and do the data support the conclusions?

Reviewer #1: Yes

Reviewer #2: Yes

3. Has the statistical analysis been performed appropriately and rigorously?

Reviewer #1: I Don't Know

Reviewer #2: Yes

4. Have the authors made all data underlying the findings in their manuscript fully available?

Reviewer #1: Yes

Reviewer #2: Yes

5. Is the manuscript presented in an intelligible fashion and written in standard English?

Reviewer #1: Yes

Reviewer #2: Yes

Reviewer #1: (No Response)

Reviewer #2: Thank you for addressing all of the reviewers' queries. The paper is much mor streamlined and clear. My only additional comment is: in line 63 of the revised ms the authors mention with certainty that the ostrich eggs have been used to store water since the MSA, however I suggest a bit more nuance, as this is an assumption but it has not been tested (e.g., no analyses of the shells have been done to determine whether they were used for water or some other liquid, or maybe they were not used as containers). So I suggest simply adding that 'it is assumed'.

**Do you want your identity to be public for this peer review?** For information about this choice, including consent withdrawal, please see our Privacy Policy

Reviewer #1: No

Reviewer #2: No

---

## [Editor Report · Acceptance letter]

PONE-D-25-36789R1

PLOS One

Dear Dr. Ferrara,

I'm pleased to inform you that your manuscript has been deemed suitable for publication in PLOS One. Congratulations! Your manuscript is now being handed over to our production team.

Kind regards,

on behalf of

Dr. Iris Groman-Yaroslavski

Academic Editor

PLOS One